# Vertebral Fractures in Acromegaly: A Systematic Review

**DOI:** 10.3390/jcm12010164

**Published:** 2022-12-25

**Authors:** Madalina Cristina Sorohan, Catalina Poiana

**Affiliations:** 1CI Parhon National Institute of Endocrinology, 011863 Bucharest, Romania; 2Department of Endocrinology, Carol Davila University of Medicine and Pharmacy, 020021 Bucharest, Romania

**Keywords:** acromegaly, vertebral fractures, osteopathy, systematic review

## Abstract

Introduction: Acromegaly is characterized by a very particular alteration of bone microarchitecture, leading to increased vertebral fragility. However, due to inconsistent and insufficient evidence, no guidelines are available for the evaluation of this osteopathy. Methods: We performed a literature review of studies published between 1968 and January 2022 on the PubMed and SCOPUS databases using the terms “acromegaly” and “vertebral fractures”. Twenty-four studies were found eligible for inclusion, published between June 2005 and November 2021. Included studies evaluated acromegaly patients, who were assessed for the presence of vertebral fractures. We excluded case reports, reviews, meta-analyses, letters to the editor, articles not written in English, and research performed on the same set of patients without significant differences in study design. Risk of bias was avoided by following the ROBIS risk of bias recommendations. We executed rigorous data collection, and the results are depicted as a narrative overview, but also, as statistical synthesis. Limitations of the evidence presented in the study include study heterogeneity, small sample sizes, and a small number of prospective studies with short follow-up. Findings: Data regarding vertebral fractures (VFs) in acromegaly and their influencing factors are variable. Twenty-four studies were included, nine out of which had a prospective design. The smallest group of acromegaly patients had 18 subjects and the largest included 248 patients. Prevalence ranges between 6.5% and 87.1%, although most studies agree that it is significantly higher than in controls. VFs also have a higher incidence (between 5.6% and 42%) and are more frequently multiple (between 46.15% and 71%). Evidence shows that disease activity and active disease duration are influencing factors for the prevalence and incidence of VFs. Nonetheless, hypogonadism does not seem to influence the frequency of VFs. While reports are conflicting regarding the use of bone mineral density in acromegaly, evidence seems to be slightly in favor of it not being associated with VFs. However, trabecular bone score is significantly lower in fractured patients, although no prospective studies are available. Interpretation: Vertebral fractures evaluation should be performed with regularity in all acromegalic patients, especially in the presence of active disease. Disease activity is an important determinant of vertebral fracture incidence and prevalence, although hypogonadism is less so. To clarify the predictive value of both BMD and TBS for vertebral fractures, additional, larger, prospective studies are necessary.

## 1. Introduction

The growth hormone/insulin-like growth factor 1 (GH/IGF-1) axis is crucial to axial growth, bone mass acquisition, and maintaining bone integrity through continuous bone remodeling [1,2,3]. Bone remodeling has two main roles, the first being to maintain normal calcium and phosphorus levels, since bones contain ~99% of total body calcium, and the second being the repair of microfractures, responding promptly to strains. The most frequent cause (>90%) of GH excess is a benign monoclonal pituitary somatotroph adenoma, uncontrollably secreting growth hormone, and leading to gigantism in childhood and acromegaly in adults [4]. Acromegaly is one of the less frequent causes of osteopathy given the rarity of the disease, with a prevalence of around 78 cases per million [5]. Under the influence of excess GH, bone remodeling is accelerated and ultimately leads to specific microarchitectural alterations of cancellous bone [6]. The clinical manifestation of this osteopathy is represented by vertebral fractures (VFs), with most studies having shown a higher prevalence than in the general population [7]. While most VFs are not clinically overt, on the long term they can lead to chronic pain, further decreasing the quality of life (QoL) for these patients [8]. Nevertheless, no guidelines for detection and prevention of osteopathy are available at present. This is mostly due to the particularity of bone changes, largely caused by microarchitectural alterations of trabecular bone and increase in cortical bone thickness [9]. Acromegalic patients seem to have normal bone mineral density (BMD), similar to that seen in healthy, age and gender-matched controls [10,11]. Thus, the use of BMD for the diagnosis of osteoporosis in these patients is less useful, at least, based on the criteria used in most of the population. Therefore, efforts have been made to find an efficient method of evaluation for these subjects. However, there is still conflicting evidence regarding the prevalence/incidence of VFs, and the usefulness of bone mineral density (BMD), trabecular bone score (TBS) and bone turnover markers. This also extends to the importance of disease activity or hypogonadism.

This is an important topic because there is a shortcoming in patient care concerning acromegalic osteopathy due to inconsistent research data. We have conducted a systematic review based on a compilation of all available research on acromegalic patients that have been evaluated for vertebral fractures in order to summarize the current state of knowledge in the field and the factors associated with the presence of VFs.

## 2. Materials and Methods

This paper was constructed according to PRISMA (The Preferred Reporting Items for Systematic reviews and Meta-Analyses, 2020) guidelines for writing systematic reviews.

Risk of bias was avoided by following the ROBIS risk of bias recommendations, a tool developed specifically for use in systematic reviews [12]. Thus, there is a three-step process that was followed in the development of this review: the assessment of relevance of each study to the intended goal, identification of concerns with the review process, and judging the risk of bias. The relevance of each study was assessed by determining whether it responded to the “target question”, which, in our case, was “Acromegaly patients evaluated for vertebral fractures”. The review process was conducted objectively and was completed by the two authors of the review, each verifying the work conducted by the other. The eligibility criteria were established beforehand and were followed by both authors. The goal was to include as many studies as possible, hence we included all studies with an acromegalic sample of patients, with an imaging evaluation of the dorso-lumbar vertebral spine, and with vertebral fractures graded according to Genant’s semiquantitative method. Excluded studies were case reports, reviews, meta-analyses, letters to the editor, articles not written in English, and research performed on the same set of patients without meaningful differences in study objectives or design (i.e, follow-ups), in which case, research with the largest cohort of patients was included.

A thorough literature search of the PubMed database was performed using the keywords “acromegaly” and “vertebral fractures”, filtered by Title/Abstract and English language, and of SCOPUS, using the same keywords, filtered by subject area for “Medicine”, document type for “Article” and language for “English”. Forty-nine search results were obtained in PubMed and thirty-nine in SCOPUS, published between December 1968 and January 2022. Assessment of titles, abstracts and full-text studies was performed for inclusion criteria. After careful consideration, 24 of these were considered eligible, published between June 2005 and November 2021. The last literature search was made in February 2022. We executed rigorous data collection and introduced this in Table 1. All statistics depicted can be found in the research of each study. The results were depicted as a narrative overview, but also, as statistical synthesis. The interpretation of findings was made according to the “target question” and elements related to it, as depicted in the tables, all performed and checked by both authors.

The following flowchart briefly depicts the screening and selection process (Figure 1).

Characteristics and main results of all studies included in the review are presented in Table 1.

## 3. Results

### 3.1. Vertebral Fractures Prevalence and Incidence

GH, as well as IGF-1, exerts direct effects on bone, supporting longitudinal bone growth as well as bone remodeling. In childhood, bone growth is accompanied by acquisition of bone mass until peak bone mass is reached, at around age 20. This process is followed by continuous and steady decrease in bone mass [3]. Whilst in physiological conditions GH and IGF-1 maintain an equilibrium in the balance between bone formation and resorption, in pathological conditions of excessive GH production, when increase in bone turnover appears, this balance is disrupted. Considering that osteoclasts take weeks to resorb a certain quantity of bone tissue while osteoblasts require months to form the same amount of bone, processes that overstimulate this mechanism will inevitably lead to bone loss [12].

Vertebral fractures are a marker of skeletal fragility, with serious consequences on both mortality and morbidity, causing increased back pain, kyphosis, limitation of daily activities, psychological distress, and even restrictive lung disease [13]. Moreover, the existence of a prevalent VF greatly increases the risk of subsequent incident VFs, by five times in the first year [14]. However, not all VFs are clinically apparent, which is why imaging techniques are crucial in evaluating patients at risk of developing VFs. The semiquantitative method of analyzing vertebrae was developed by Genant in 1996 and is now the most widely used technique of identifying and classifying VFs. The fractures are classified according to severity as mild (reduction in height between 20–25%), moderate (reduction between 25–40%), and severe (reduction of >40%) [15]. The current management guidelines for acromegaly have no recommendations for the evaluation of bone involvement [16,17]. 

The study of Bonadonna et al. [18] was the first to report prevalent VFs in acromegaly, with a frequency of 52.8% in a cohort of 36 postmenopausal acromegalic patients. They included all fractures that were observed on x-ray scans, from mild to severe, 78.9% of these being mild and only 5.3% severe. Since then, subsequent papers have shown varying results on prevalent VFs, from 6.5% [19] to 87.1% [20]. This inconsistency could be due to the variable types of VFs reported, the number of patients enrolled, and examiner variability. A total of 22 studies have reported VFs of all grades [18,19,20,21,22,23,24,25,26,27,28,29,30,31,32,33,34,35,36,37,38,39], three out of which have only identified mild and moderate fractures [33,34,35]. The prevalence varied from 10.6% to 23.68% in the studies that only reported moderate and severe VFs [30,40,41]. Moreover, between 46.15% and 71% of VFs were observed to be multiple [18,21,24,30,36]. The largest study was that of Mazziotti et al. [37], who performed a multicentric research on 248 acromegalic patients. They observed a prevalence of 31.45% and an incidence of 26.21% for VFs during the 48 months of follow-up. Eight other studies had a prospective design, with a follow-up period between 24 and 108 months [19,20,22,23,32,38,39], with the exception of Sala et al., who reevaluated patients at 12 months after cure or control of the disease [33]. In this period, incident VFs were observed in 5.6% to 42% of patients.

### 3.2. Disease Activity

Around 70% of somatotroph adenomas are diagnosed as macroadenomas, which decreases the chance of obtaining a surgical cure [42]. In centers with vast experience in transsphenoidal resection of pituitary adenomas, the cure rates are >80% for microadenomas and <50% for macroadenomas [43,44,45], with an overall long-term cure of around 60% [46]. Furthermore, the cure rate after stereotactic radiosurgery is around 50% at 7 years [46,47]. The definition of disease control, according to the guidelines elaborated in 2014, co-sponsored by the European Society of Endocrinology, is represented by normal serum IGF-1 for sex and age and random GH < 1.0 µg/L. Patients with persistent disease after surgical intervention are usually put under medical therapy with either a somatostatin receptor ligand (SRL) or Pegvisomant [16,17]. Disease control is obtained in 20–60% of patients with the use of first-generation SRL [48]. Pasireotide, a second-generation SRL, has shown to be superior to Octreotide in achieving biochemical disease control [49]. The efficiency of Pegvisomant was around 60–70% in “Acrostudy” [50] but much greater in studies that used higher doses, normal IGF-1 being obtained in 90–97% of patients [51,52]. Alternative treatments include combination therapy between SRL, Pegvisomant or Cabergoline (dopamine agonist) [53]. Taking all of this into account, we have to consider the effect disease activity has on the prevalence and incidence of VFs. Three of the studies only analyzed patients with controlled disease [20,26,28], another three only enrolled patients with active disease [33,38,39] and among the others, the frequency of uncontrolled/active disease varied between 29.8% and 77.1%. Six of the studies analyzed the association between disease activity and the prevalence or incidence of VFs. Bonadonna et al. [18] and Uygur et al. [29] observed the association between VF prevalence and active disease, the first found an association between the two but the second did not. Pelsma et al. [20] found an association between active disease duration and VF prevalence, but not with VF progression. Incident VFs were associated with both active disease and active disease duration in the research of Mazziotti et al. [37]. Chiloiro at al found an association between incident VFs and persistence of active acromegaly at follow-up [38], and in another study, they also found an association with active disease duration [39]. Regarding the prevalence of VFs in patients with controlled disease, in the three studies that only analyzed patients with controlled disease, it ranged from 11.53% in the study of Calatayud et al. [29] to 59% in that of Wassenaar et al. [27] and 87.1% in that of Pelsma et al. [21]. The latter also found an incidence of 35.5% of VFs during a follow-up period of 9.1 years [21]. Bonadonna et al. found a similar prevalence in patients with controlled disease as in the control postmenopausal group, of 33.3% [19]. Uygur et al. reported a prevalence of 22.2%, similar to that of patients with active disease, and Mazziotti et al., in the prospective study from 2020, documented an incidence of 18.24% during their follow-up period of four years, compared to 36.03% in patients with active disease [38]. Interestingly, in their 36-month follow-up, Chiloiro et al. [39] observed significantly fewer incident VFs in patients treated with Pasireotide compared to Pegvisomant, with both active and controlled disease at the end of the study. All available details are presented in Table 2.

### 3.3. Hypogonadism

Maintaining a normal remodeling process within the basic multicellular units (BMUs) is a complex task undertaken by many hormones, proteins, growth factors and cytokines [54]. Estrogen has been repeatedly shown to be a major contributor to maintaining a normal bone metabolism. Its effects present mainly on osteoclasts but also on osteocytes. Both in vitro and in vivo studies have shown that estrogen suppresses osteoclast production and differentiation, and increases apoptosis through direct as well as indirect effects, via RANKL [55,56,57,58,59]. Another target for estrogen’s actions is the osteoclast, the mechanosensor for bone stress and injury [60]. Studies have shown that inducing estrogen deficit leads to a 4-fold increase in osteocyte apoptosis in both cortical and trabecular bone [61,62,63], also subsequently leading to an increase in RANKL levels in the microenvironment of the bone [64,65]. These result in an increase in bone resorption. Last but not least, estrogen acts on osteoblasts by decreasing osteoblast apoptosis, thus increasing their lifespan [66]. These processes occur not only in women but also in men, estrogen having a more important role in regulating bone metabolism than testosterone [67]. Twelve of the twenty-four studies have analyzed the correlation between VFs and hypogonadism in acromegaly. Four of these found an association between them, two of which referred to prevalent VFs and two referred to incident VFs [19,21,23,29]. The other eight did not find this association, five of them referring to prevalent VFs, two referring to incident VFs and one to VF progression [20,24,25,27,32,35,38,39]. Available data is presented in the table below (Table 3).

### 3.4. Bone Mineral Density

Measurement of bone mineral density, with the help of dual-energy x-ray absorptiometry (DXA), at the lumbar spine, femoral neck, total hip, and at the distal radius is the current gold standard for the diagnosis and monitoring of patients at risk for osteoporosis and osteoporotic fractures [68]. The World Health Organization (WHO) criteria for the diagnosis of osteoporosis is based on the T score deviation compared to the mean peak bone mass of healthy adults, a normal T score being between +1 and −1 standard deviations (SD), with osteopenia being defined as a T score between −1 and −2.5 SD, and osteoporosis below −2.5 SD [69]. The Z score compares the patient’s BMD with the mean predicted BMD from an age-matched population [70]. The T score is used for the diagnosis of osteoporosis in postmenopausal women and men over 50 years of age and for all other categories, the Z score is used. The most frequent cause of osteoporosis is the onset of menopause; therefore, extensive guidelines have been drawn up for the management of postmenopausal osteoporosis. However, other causes of bone fragility lack clear recommendations on this issue. The use of BMD in the evaluation of acromegalic patients has always been controversial. Eleven studies have compared acromegalic patients with healthy controls, although only eight of them compared BMD between the two groups of patients. Five studies found no difference in the comparison [23,25,28,31,33]: Bonadonna et al. found higher lumbar spine T score in acromegalic patients [18], Uygur et al. found a higher BMD at the femoral neck in the acromegalic group [29], and Kužma et al. found a lower lumbar spine areal BMD (aBMD) and lower trabecular volumetric BMD (vBMD) at the femoral neck and total hip by using a 3D Shaper software (version 2.7, Galgo Medical S.L, Barcelona, Spain) [36]. The association between VFs and BMD has also been largely debated, with contradictory results. A total of 17 studies have assessed this association, nine of them finding no connection [21,22,24,25,26,29,38,39,41]. Among these, Chiloiro et al. has reported the connection between incident VFs and BMD in both studies. The remaining eight that did find this association also had discrepant results. Kužma et al. found in their first study from 2019 that acromegalic patients had lower aBMD at the femoral neck and, also, lower total hip cortical sBMD and total hip cortical vBMD; furthermore, in 2021 they reported the association between VF prevalence and change in BMD at follow-up as a significant decrease in the trabecular, cortical and surface BMD at the femoral neck and total hip in fractured patients [32,36]. Silva et al. found a significantly higher lumbar spine and distal radius Z score in fractured patients [31], Carbonare et al. had similar results, having also found higher lumbar spine BMD and Z score in fractured subjects [27]. On the other hand, Plard et al. found a significantly lower T score at the lumbar spine in fractured subjects [34] and Bonadonna et al. also observed lower lumbar spine BMD in these patients [18]. Mazziotti et al. in their study from 2013 detected lower femoral neck BMD in subjects with incident VFs and controlled disease [23]. Likewise, Battista et al. found the mean of Z single-energy quantitative computed tomography (Z-QCT) values measured at baseline and at end of follow-up (M Z-QCT) to be a predictive factor for incident VFs [19]. Details are depicted in Table 4 (where multiple BMD were provided, either the statistically significant one or the BMD from the lumbar spine was provided).

### 3.5. Trabecular Bone Score

TBS is an index obtained by analyzing the amplitude of gray-level pixel variations on the lumbar spine DXA scan. Its results correlate with bone volume fraction and mean bone thickness, providing an indirect view of bone microarchitecture [71]. There is no general agreement for what TBS values are considered normal or not, and so the cut-off points proposed by the manufacturer are the ones currently being used: values ≥ 1.350 being normal, between 1.200 and 1.350 constituting partially degraded bone, and ≤1.200, degraded bone [72]. There are still no recommendations for the use of TBS in clinical practice. Some categories of patients seem to benefit more from the use of this index, mainly those that are known to have an increased risk of osteoporotic fractures but in whom the use of the Fracture Risk Assessment Tool (FRAX) or BMD is inefficient. The most popular use is probably in patients with type 2 diabetes mellitus, in whom BMD does not correlate with fracture risk, TBS being able to independently predict incident osteoporotic fractures [73]. Other possible uses could be in patients with glucocorticoid excess, primary hyperparathyroidism, chronic kidney disease, rheumatoid arthritis or thalassemia major [74,75,76,77,78,79,80]. Its usefulness in the evaluation of acromegalic osteopathy is not yet clear. Only a few studies have assessed bone quality in acromegaly with the use of TBS, however, all of these have found significantly lower TBS values compared to healthy controls [10,11,28,33,36]. Three studies have investigated the link between TBS and prevalent VFs, but no study has yet examined the connection with incident VFs. Kužma et al. in their study from 2019 found significantly lower TBS values in fractured acromegalic subjects [36], as did Calatayud et al. [28]. As a follow-up to their first study, Kužma et al. [32] reported in 2021 a significant association between prevalent VFs and the change in TBS from baseline to 24 months in controlled patients. Data is shown in Table 5.

### 3.6. Bone Turnover Markers

The process of bone remodeling taking place in BMUs consists of a balance between bone resorption and bone formation, undertaken by osteoclasts and osteoblasts, respectively. Bone turnover markers (BTM) can be used to assess bone metabolism, being either bone matrix components or enzymes reflecting the metabolism of osteoblasts and osteoclasts [81,82]. The prime marker for bone resorption is serum C-terminal cross-linking telopeptide of type I collagen (CTX), a product of the resorption of bone collagen by osteoclasts. CTX levels are influenced by both food ingestion and time of day, hence samples must be collected in the morning after a night of fasting, when levels are highest [83]. As for bone formation marker representatives, osteocalcin and procollagen type 1 N-terminal propeptide (P1NP) are archetypal. Osteocalcin is a protein synthesized by osteoblasts under the influence of 1,25-dihydroxyvitamin D. Osteocalcin is transformed into its mature form by γ-glutamyl carboxylase (GGCX) at positions 17, 21 and 24. This carboxylation raises osteocalcin’s affinity for hydroxyapatite crystals, thus attaching it to the bone matrix. On the other hand, bone resorption leads to the de-carboxylation of osteocalcin, causing the release of under-carboxylated osteocalcin into the circulation [84,85]. P1NP results from the cleavage of procollagen in the process of type 1 collagen formation, predominantly originating from bone but also, in a small proportion, from cartilage, tendons and skin. The original form is an unstable trimeric form, which converts to dimeric and monomeric forms. The assay is very stable and has a low variability [86]. Sclerostin is a more novel discovery in terms of bone resorption markers, acting as an inhibitor of Wnt/βcatenin signaling in bone. The Wnt signaling pathway promotes osteoblastic differentiation and maturation, increases osteoblastic and osteocytic survival, and decreases osteoclastic formation through elevated osteoprotegerin production [87]. Other BTM include bone-specific alkaline phosphatase (BSALP), procollagen type 1 C-terminal propeptide (P1CP), tartrate-resistant acid phosphatase (TRACP), pyridinoline (PYD), deoxypyridinoline (DPD), hydroxyproline (OHP), osteoprotegerin (OPG), galactosyl hydroxylysine (GHL), bone sialoprotein (BSP), and Cathepsin K [81,82,88]. Nonetheless, not all of these are commonly used in clinical practice. Nine studies have evaluated the association between VFs prevalence or incidence and bone turnover markers. Kužma et al. in 2021 [32] found a significant change in CTX levels between baseline and follow-up in fractured patients, while in their study for 2019 [36], they found significantly lower P1NP levels in the presence of VFs. Bonadonna et al. also found higher levels of BSALP in fractured subjects [18]. The other six studies found no association between BTM and prevalent or incident VFs [20,22,26,27,29,31].

## 4. Study Limitations

Limitations of the evidence presented in the study include study heterogeneity, small sample sizes, small number of prospective studies with short follow-up. We also limited publications to English language. However, given the rarity of the disease, we consider these observations to be relevant and meaningful, and to provide an insight into the specifics of bone disease in acromegaly.

## 5. Conclusions

Acromegaly is a known cause of skeletal fragility, characterized mainly by vertebral fragility. Despite the large variability of data regarding vertebral fracture prevalence and incidence, both are clearly substantially higher in this pathology than in the general population. Multiple vertebral fractures are also more frequent. There are multiple factors to consider that influence the frequency of vertebral fractures, such as disease activity, disease duration and associated hypogonadism. Patients with active disease have a higher prevalence of vertebral fractures and incident fractures, correlating with both active disease and duration of active disease. The evidence concerning the influence of hypogonadism on vertebral fractures is contradictory, although leans towards this not being a risk factor, neither for prevalent nor for incident fractures. Possibly the most controversial subject of discussion in these patients is the means of evaluation of bone density of microarchitecture. While most studies agree that acromegalic subjects have similar BMD to healthy controls and do not usually meet the criteria for osteoporosis, the evidence on the association of fractures and BMD is less universal. Almost half of the studies did not find any association, while among the rest, some found higher lumbar spine BMD, and others, lower lumbar spine or femoral neck BMD. It is safe to say that, on this subject, no consensus has been reached. TBS is lower in acromegalic patients than in controls and was also observed to be significantly lower in fractured patients compared to non-fractured patients. Nonetheless, these observations were made on prevalent vertebral fractures in just three studies, thus need to be validated in further, prospective studies. So far, no specific bone turnover marker has shown consistency in associating with vertebral fractures in acromegaly. Larger prospective studies should be performed in order to clarify the predictive value of both BMD and TBS for vertebral fractures, as current evidence cannot support the use of either in acromegaly. Findings from this review suggest that there is a strong recommendation for systematic, periodic assessment of vertebral fractures in acromegaly, regardless of disease activity or presence of hypogonadism, but especially in patients with uncontrolled disease or prevalent vertebral fractures. Contributing with data on the use of BMD, TBS or other imaging techniques from larger prospective studies could help in implementing clear guidelines to serve, not only in diagnosing, but also in preventing incident vertebral fractures in acromegaly. It could also be worth evaluating the impact that preventive anti-osteoporotic treatment could have on incident vertebral fractures, especially in high-risk patients.

## Figures and Tables

**Figure 1 jcm-12-00164-f001:**
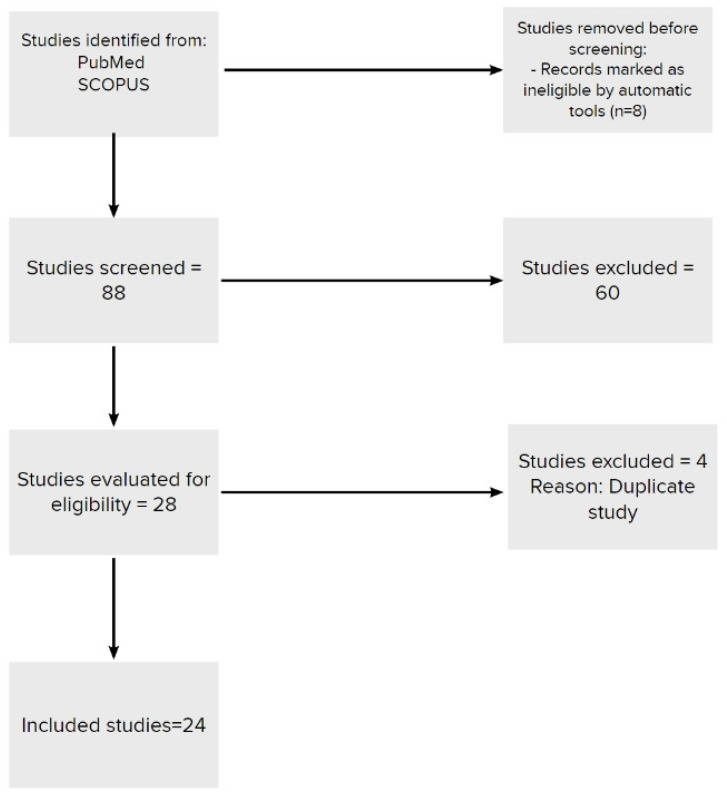
Flowchart of the screening and selection process.

**Table 1 jcm-12-00164-t001:** Summary of studies assessing vertebral fractures in acromegaly patients.

Authors, Year, Country	Study Design	Sample (*n*)	Disease Activity	Follow-Up	Main Results					
					VFs	+VFs and BMD	+VFs and TBS	+VFs and BTM	+VFs and Hypogonadism	Others
Kužma et al., 2021 [1]	Prospective	70 ACM	Active: 37.14%	24 months	Prevalence: 10% ** Incidence: 8.57% **	Yes ^◊^ (change at follow-up at trabecular, cortical and surface FN and TH BMD)	Yes (chance at follow-up)	Yes (change in CTX at follow-up)	No	
Silva et al., 2021 [2]	Cross-sectional	30 ACM 53 controls	Active: 40%	N/A	Prevalence: 26.66% **	Yes ↑ LS Z score ↑ 1/3 R Z score	N/A	No	N/A	HR-pQCT parameters did not differ according to VF presence
Sala et al., 2021 [3]	Cross-sectional & prospective	18 ACM 36 controls	Newly diagnosed/Active	Baseline & 12 months after cure/control	Prevalence: 11% * Incidence: 5.6%	N/A	N/A	N/A	N/A	
Calatayud et al., 2021 [4]	Cross-sectional	26 ACM 117 controls	Controlled	N/A	Prevalence:11.53% **	N/A	Yes ↓TBS	N/A	N/A	
Uygur et al., 2021 [5]	Cross-sectional	70 ACM 70 controls	Active: 77.1%	N/A	Prevalence:72.9% **	No	N/A	No	Yes	Active vs. controlled disease had no differences in VF prevalence
Pelsma et al., 2020 [6]	Prospective	31 ACM	Controlled	Baseline & 9.1 years	Prevalence: 87.1% ** Incidence: 35.5% **	N/A	N/A	No	No (progression of VF)	Prevalence of VF associated with active disease duration Progression of VF not associated with active disease duration
Cellini et al., 2020 [7]	Cross-sectional	38 ACM 38 controls	Active: 29.8%	N/A	All: 34.21% ** Moderate/severe: 23.68% Multiple: 61.53% **	N/A	N/A	N/A	N/A	VF associated with higher spinal imbalance, ↓ AcroQoL and SF36 general health scores and ↑ WOMAC pain and WOMAC global scores
Mazziotti et al., 2020 [8]	Retrospective, Multicentric	248 ACM at baseline; 52 received bone active drugs	Active:44.75%	Baseline & 48 months	Prevalence:31.45% ** Incidence: 26.21% **	N/A	N/A	N/A	N/A	Incident VF associated with prevalent VF, active acromegaly, duration of active acromegaly and treated hypoadrenalism
Plard et al., 2020 [9]	Cross-sectional, monocentric	50 ACM	N/A	N/A	Prevalence: 6% *	Yes ↓ LS T score	N/A	N/A	N/A	
Chiloiro, 2020 [10]	Retrospective, multicentric	55 ACM	Active	36 months	Prevalence: 41.8% ** Incidence: 29.1% **	No (incident VFs)	N/A	N/A	No (incident VFs)	Pasireotide vs. Pegvisomant had ↓ incident VFs; Incident VFs associated with prevalent VFs, persistence of active acromegaly and higher IGF-1 at study end
Oliveira et al., 2019 [11]	Prospectively recruited, monocentric	58 ACM	Active: 30.4%	N/A	Prevalence: 13.8% *	N/A	N/A	N/A	No	
Pontes et al., 2019 [12]	Prospectively recruited, monocentric	89 ACM	Active: 63%	N/A	Prevalence: 14%	N/A	N/A	N/A	N/A	No difference in VF frequency between patients with fl-GHR and those with d3-GHR
Kužma et al., 2019 [13]	Cross-sectional, monocentric	106 ACM 104 controls	Active: 33%	N/A	Prevalence: 12% ** Multiple: 46.15%	Yes ^◊^ ↓ FN aBMD ↓ TH cortical sBMD ↓ TH cortical vBMD	Yes ↓TBS	Yes ↓ P1NP	N/A	
Chiloiro et al., 2019 [14]	Retrospective, longitudinal, multicentric	83 ACM	Active	82 months	Prevalence: 28.9% ** Incidence: 34.9% **	No (incident VFs)	N/A	N/A	No (incident VFs)	Incident VFs associated with prevalent VFs, duration of active acromegaly and IGF-I during follow-up
Carbonare et al., 2018 [15]	Cross-sectional, monocentric	47 ACM	Active: 36.17%	N/A	Prevalence: 63.8% **	Yes ↑ LS BMD and Z	N/A	No	No	ACM vs. controls: ↓ bone volume/tissue volume, ↓ trabecular thickness and ↑ trabecular separation.
Maffezzoni et al., 2016 [16]	Cross-sectional, monocentric	40 ACM 21 Controls	Active: 50%	N/A	Prevalence: 37.5% ** Multiple: 66.66%	No	N/A	N/A	Yes ***	HR-CBCT evaluation: Fractured ACM had lower BV/ TV, greater Sp.mean, higher cortical porosity
Mormando et al., 2014 [17]	Cross-sectional, monocentric	109 ACM	Active:33%	36 months	Prevalence: 43.1% ** Incidence: 37.1% **	No	N/A	No	N/A	-VF prevalence ↑ in d3-GHR carriers than fl-GHR patients -d3-GHR carriers were 3 times more likely to suffer a VF.
Mazziotti et al., 2013 [18]	Prospective, multicentric	88 ACM 106 Controls	Active:45.5%	36 months	Prevalence: 38.6% ** Incidence: 42% **	Yes ↓ FN BMD (incident VFs in controlled patients)	N/A	N/A	Yes (incident VFs in controlled patients)	
Madeira et al., 2012 [19]	Cross-sectional, monocentric	75 ACM	Active: 72%	N/A	Prevalence: 10.6%	No	N/A	N/A	N/A	
Padova et al., 2011 [20]	Retrospective	20 ACM	Active: 60%	N/A	Prevalence: 39% ** Multiple: 71%	No	N/A	N/A	No	
Mazziotti et al., 2011 [21]	Cross-sectional	57 ACM 57 Controls	Active: 36.8%	N/A	Prevalence: 50.9% **	No	N/A	N/A	No	Diabetic patients had ↑ prevalence of VFs
Wassenaar et al., 2011 [22]	Cross-sectional, monocentric	89 ACM 3469 Controls	Controlled	N/A	Prevalence: 59% **	No	N/A	No	N/A	Hypogonadal men had a higher prevalence of VFs Men had a higher risk of developing VFs
Battista et al., 2008 [23]	Longitudinal, retrospective	46 ACM	Active:47.82%	48 months	Prevalence:6.5% ** Incidence: 28.26% **	Yes (incident VFs)	N/A	N/A	Yes (incident VFs)	VFs incidence was associated with hypogonadism and M Z-QCT (L1-L4)
Bonadonna et al., 2005 [24]	Cross-sectional	36 ACM 36 Controls	Active:41.66%	N/A	Prevalence: 52.8% ** ^ Multiple: 68.4%	Yes ↓ LS BMD	N/A	Yes ↑ BSALP	N/A	VFs prevalence was associated with active disease

* Fractures were mild (grade 1) and moderate (grade 2). ** Fractures were mild (grade 1), moderate (grade 2) and severe (grade 3). *** Refers to untreated hypogonadism. ^ Bonadonna et al. [24] used the 1996 Genant classification of VF (mild 20–25%, moderate 25–35%, severe > 35%). ^◊^ Kužma et al. [1] performed BMD evaluation using both DXA and a 3D Shaper software. ↑: increased, ↓: decreased, ACM: acromegaly; VFs: vertebral fractures, BMD: bone mineral density, TBS: trabecular bone score; BTM: bone turnover markers; N/A: not applicable; HR-CBCT: high resolution cone-beam computed tomography; BV/TV: bone volume/trabecular volume ratio, Sp.mean: mean trabecular separation; fl-GHR: full-length isoform of growth hormone receptor; d3-GHR: exon 3-deficient isoform of growth hormone receptor; QCT: single-energy quantitative computed tomography; BSALP: bone-specific alkaline phosphatase; AcroQoL: Acromegaly quality of life Questionnaire; SF-36: Short Form-36; WOMAC: Western Ontario and Mc Master University Osteoarthritis Index; 1/3 R—1/3 radius; M Z-QCT: mean of Z-QCT values measured at baseline and at end of follow-up.

**Table 2 jcm-12-00164-t002:** Correlations reported between disease activity and prevalence/incidence of vertebral fractures.

Study	No. of Patients	VF Prevalence		*p*	Active Acromegaly		*p*
		Active Disease	Controlled Disease		With VFs	No VFs	
Silva et al., 2021 [2]	30				62.5%	31.6%	0.206
Cellini et al., 2020 [7]	38				23.1%	32%	0.71
Mazziotti et al., 2020 [8]	248				61.54% ^#^ 23.08% ^@^	38.8% ^#^ 13.11% ^@^	**0.002** 0.058
Chiloiro, 2020 [10]	55				52.9% ^@^	47.1% ^@^	**0.009**
Kužma et al., 2019 [13]	106	11.42%	12.67%	N/S			
Chiloiro et al., 2019 [14]	83				OR 4.01 (0.29–56.12) ^!^		0.30
Carbonare et al., 2018 [15]	47	76%	57%	0.31			
Mazziotti et al., 2013 [18]	88				67.6% ^#^ 32.4% ^@^	29.4%^#^ 3.9%^@^	**<0.001** **<0.001**
Mazziotti et al., 2011 [21]	57				53.3%	18.5%	**0.007**
Battista et al., 2008 [23]	46	27.27%	29.16%	0.61			
Bonadonna et al., 2005 [24]	36	80%	33.3%	**0.008**			

N/S—not significant; ^#^—referring to incident VFs associated with active acromegaly at study entry; ^@^—referring to incident VFs associated with active acromegaly at follow-up; ^!^—Odds ratio for incident vertebral fractures; OR—odds ratio; *p* values with bold are statistically significant (*p* < 0.05).

**Table 3 jcm-12-00164-t003:** Correlations reported between the presence of hypogonadism and vertebral fractures prevalence/incidence.

Study	No. of Patients	VFs Prevalence		*p*	Hypogonadism & Incident VFs		*p*	Hypogonadism & Prevalent VFs		*p*
		Eugonadism	Hypogonadism		Yes	No		Yes	No	
Uygur et al., 2021 [5]	70	87.1%	61.5%	**0.017**				52.94%	7.84%	**0.01**
Mazziotti et al., 2020 [8]	248				69.23%	60.66%	0.21			
Kužma et al., 2019 [13]	106	92.3%	7.69%	N/S						
Chiloiro et al., 2019 [14]	83				1.44 (0.44–4.73) ^		0.55			
Maffezzoni et al., 2016 [16]	40							73.3%	32%	**0.01**
Mormando et al., 2014 [17]	109	48.93%	51.06%	N/P						
Mazziotti et al., 2013 [18]	88				35.2%	23.5%	0.23			
Mazziotti et al., 2011 [21]	57	30%	14.8%	0.17				30%	14.8%	0.17
Battista et al., 2008 [23]	46	17.39% ^€^	39.13% ^€^	**0.02**						

N/S—not significant; ^—Odds ratio for incident VFs; €—refers to incidence of VFs in hypogonadal vs. eugonadal patients, *p* values with bold are statistically significant (*p* < 0.05)

**Table 4 jcm-12-00164-t004:** Correlations reported between bone mineral density and vertebral fracture prevalence/incidence.

Study	No. of Patients	BMD		*p*	BMD		*p*
		With Prevalent VFs	No Prevalent VFs		With Incident VFs	No Incident VFs	
Kužma et al., 2021 [1]	70	–1.71 ^β^	1.79 ^β^	**<0.05**			
Silva et al., 2021 [2]	30	1.228 ± 0.056 ^α^	1.159 ± 0.161 ^α^	0.116			
Uygur et al., 2021 [5]	70	1.218 ± 0.220	1.199 ± 0.187	0.76			
Chiloiro, 2020 [10]	55				0.1 (1.7) ^$^	−1.05 (2.4) ^$^	0.9
Kužma et al., 2019 [13]	106	0.74 ± 0.14 ^£^	0.82 ± 0.12 ^£^	**≤0.05**			
Chiloiro et al., 2019 [14]	83				−0.12 (−2.80−+3.51) ^$^	−0.69 (−2.90−+ 1.60) ^$^	0.12
Carbonare et al., 2018 [15]	47	1.07 (0.84–1.47)	0.96 (0.82–1.34)	**0.021**			
Maffezzoni et al., 2016 [16]	40	+0.35 (−2.8–+4.4) ^$^	+0.01 (−3.2–+4.9) ^$^	0.45			
Mormando et al., 2014 [17]	109	1.112 (±0.165)	1.158 (±0.167)	0.210			
Mazziotti et al., 2013 [18]	88				+2.5 (−8.8–+15.3) ^&^	+0.1 (−18.9–+11.5) ^&^	0.07
Madeira et al., 2012 [19]	75	0.887 ± 1.026	1.181 ± 0.213	0.93			
Padova et al., 2011 [20]	20	0.97 ± 0.12	0.94 ± 0.17	N/P			
Mazziotti et al., 2011 [21]	57	+0.1 (−2.6–+2.8) ^$^	−1.0 (−2.9–+2.6) ^$^	0.47			
Wassenaar et al., 2011 [22]	89	1.00 (0.03)	1.06 (0.04)	0.29			
Battista et al., 2008 [23]	46				N/P	N/P	**0.035 ****
Bonadonna et al., 2005 [24]	36	−1.7 (−3.4–+1.3) *	−0.8 (−4.1–+1.3) *	**0.05**			

*—Results expressed as T score; ** M Z-QCT associated with incident VFs; ^$^—Lumbar spine BMD Z score; ^&^—change in lumbar BMD (%); £—Neck areal BMD; ^α^—aBMD, areal bone mineral density at the lumbar spine; ^β^—change in cortical vBMD (mg/cm^3^) at 24-month follow-up compared to baseline; N/P-not provided, *p* values with bold are statistically significant (*p* < 0.05)

**Table 5 jcm-12-00164-t005:** Correlations reported between trabecular bone score and vertebral fractures prevalence.

Study	Number of Patients	TBS		*p*
		With Prevalent VFs	No Prevalent VFs	
Kužma et al., 2021 [1]	70	–1.4 ^	+1.3 ^	**<0.05**
Calatayud et al., 2021 [4]	26	1.13 ± 0.07	1.29 ± 0.13	**0.03**
Kužma et al., 2019 [13]	106	1.118 ± 0.12	1.202 ± 0.13	**≤0.001**

^—comparison of change in TBS at 24 months compared to baseline, *p* values with bold are statistically significant (*p* < 0.05)

## Data Availability

Data is available in the studies included.

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
