# Peer review of "Vertebral Fractures in Acromegaly: A Systematic Review"

_jcm, 2022, doi:10.3390/jcm12010164_

Round 1

Reviewer 1 Report

Major observation

As data are unavailable for most parameters under the Main Results of Table 1, this systematic review seems premature.

Minor observations

Mentioning authors’ gender should be avoided throughout the text, such as that mentioned in line 117 for Mazziotti et al.

A separate section on the strengths and weaknesses of this review should be described.

Author Response

Dear reviewer,

Thank you for your suggestions.

Concerning your observation, data noted as “N/A” signify “not applicable”, not “not available”.  As we have mentioned in the “Methods” section, we included all studies that have analyzed the presence of fragility vertebral fractures in patients with acromegaly. The studies included did not have the same design and analyzed the association between vertebral fractures and different parameters (BMD, TBS, bone turnover markers, disease activity, hypogonadism).

Regarding your other observation, we have corrected any mention of the authors’ gender.

Reviewer 2 Report

In the manuscript " Vertebral Fractures in Acromegaly: A systematic Review", the authors present a meta-analysis of the presence of vertebral fractures in patients with active or treated acromegaly. The paper add value to the literature and present a very interesting review of the literature.

the paper is well written and present well the pathophysiological process of the vertebral fractures in patients with acromegaly, as well as a clear an objective evaluation of the available papers in the literature. 

An interesting thing would be to see if the vertebral fractures are still prevalent in patients with treated acromegaly and if there any other factors ( such as treatement or comorbidities) that can also contribute or influence the general outcome.

Author Response

Dear reviewer,

Thank you very much for your review.

We added in the paragraph entitled “Disease activity” details regarding the prevalence of vertebral fractures in patients with controlled acromegaly. As we had previously mentioned in the paragraph, only six of the studies included analyzed the prevalence of vertebral fractures taking into consideration disease activity. So, we detailed the prevalence of VFs in these studies and, also, in the studies that included only patients with controlled disease. The added sentences are as follows and you can find the highlighted in the manuscript.

“Regarding the prevalence of VFs in patients with controlled disease, in the three studies that only analyzed patients with controlled disease, it ranged from 11.53% in the study of Calatayud et al[28] to 59% in that of Wassenaar et al[26] to 87.1% in that of Pelsma et al[20]. The latter also found an incidence of 35.5% of VFs during a follow-up period of 9.1 years[20]. Bonadonna et al found a similar prevalence in patients with controlled disease as in the control postmenopausal group, of 33.3%[18]. Uygur et al reported a prevalence of 22.2%, similar to that of patients with active disease, and Mazziotti et al, in the prospective study from 2020, documented an incidence of 18.24% during their follow-up period of four years, compared to 36.03% in patients with active disease[37]. Interestingly, Chiloiro et al observed in their 36 month of follow-up significantly fewer incidental VFs in patients treated with Pasireotide compared to Pegvisomant, with both active and controlled disease at the end of the study[38]”.

Reviewer 3 Report

Dear authors,

Your systematic review covers very interesting and important issue in patients with acromegaly. However, there is need for improvement of data presentation.

First, you should improve Abstract according to PRISMA guidelines. There is also need for improvement in English.

In Materials and Methods:

Please enter Diagram for study selection. Also please, enter methods for study risk of bias assessment. Please describe the process used to decide which study was eligible.

Results: Currently, your results are presented more as discussion than the results of systematic review. Please, for the Result there is need to follow PRISMA guidelines.

Results should have:

For outcome (VF), for each study: (a) summary statistics for each group (where appropriate) and (b) an effect estimate and its precision (e.g. confidence/credible interval), ideally using structured tables or plots.

For each synthesis, briefly summarize the characteristics and risk of bias among contributing studies. Present results of all statistical syntheses conducted. 

Present results of all investigations of possible causes of heterogeneity among study results.

Present results of all sensitivity analyses conducted to assess the robustness of the synthesized results.

Present assessments of risk of bias due to missing results (arising from reporting biases) for each synthesis assessed.

Present assessments of certainty (or confidence) in the body of evidence for each outcome assessed.

For the Discussion:

Provide a general interpretation of the results in the context of other evidence.

Discuss any limitations of the evidence included in the review.

Discuss any limitations of the review processes used.

Discuss implications of the results for practice, policy, and future research.

Author Response

Dear reviewer,

We would like to thank you for your recommendations. We have tried to add as much of the suggestions as possible and we are certain they will improve the quality of the manuscript.

You will find all the modifications highlighted in the text and abstract, as well as the flowchart of the selection process and additional tables that include the available statistics from the studies.

We hope the alterations meet your expectations and look forward to your response.

Round 2

Reviewer 1 Report

The revised manuscript has been improved

Author Response

Thank you!

Reviewer 3 Report

Dear authors,

Thank you for the changes. The manuscript was significantly improved. There is one minor mistake to correct. Please, at Table 2. there is missing p value for the study Kužma et al,2019. After this correction the manuscript could be accepted for publication.

Author Response

Dear reviewer,

Thank you for your response.

Concerning the p value from Table 2 for the study of Kuzma et al from 2019, the authors did not provide all the p values at the comparison of variables, they just marked the ones that were statistically significant. And the p in this case was not statistically significant. So we have written down “N/S” in the table, for not significant.